# HIV-Infected Women with Low-Grade Squamous Intraepithelial Lesion on Cervical Cytology Have Higher Risk of Underlying High-Grade Cervical Intraepithelial Neoplasia

**DOI:** 10.3390/ijerph181910211

**Published:** 2021-09-28

**Authors:** Nawin Sakdadech, Tanarat Muangmool, Jatupol Srisomboon

**Affiliations:** Department of Obstetrics and Gynecology, Faculty of Medicine, Chiang Mai University, Chiang Mai 50200, Thailand; nawin_101@hotmail.com (N.S.); tanarat.m@cmu.ac.th (T.M.)

**Keywords:** LSIL, cervical cytology, HIV, uterine cervix, cervical screening result

## Abstract

Objective: To evaluate the risk of histological high-grade cervical lesions defined as cervical intraepithelial neoplasia grade 2 or worse (CIN2+) in women with human immunodeficiency virus (HIV) infection who had low-grade squamous intraepithelial lesions (LSIL) on cervical cytological screening compared with HIV-uninfected women who had similar cytology. Methods: 127 HIV-positive women aged 18–65 years with LSIL cytology undergoing colposcopic examination between January 2008 and December 2019 at Chiang Mai University Hospital were reviewed. By matching 1:1 ratio for age (±5 years) and examination time period (±12 months), 127 HIV-negative women with LSIL cytology in the same period were recruited as controlled subjects for comparison. The patients’ characteristics, HIV status, CD4 counts, antiretroviral therapy, and histopathology on cervical biopsy were analyzed. Results: HIV-infected women significantly had early sexual debut (age < 20 years) and more sexual partners (≥2) than HIV-uninfected women. The risk of underlying CIN2+ in HIV-infected women was significantly higher than that in HIV-negative women (20.5% vs. 9.4%, *p* = 0.021) with an odds ratio (OR) of 2.47 and 95% confidence interval (CI) = 1.18–5.14. After adjustment, the risk of underlying CIN2+ in HIV-infected women remained significantly higher than that in HIV-uninfected women (adjusted OR = 2.55, 95% CI = 1.11–5.82, *p* = 0.027). Conclusion: Among women with LSIL on cervical cytology, the risk of underlying CIN2+ in HIV-infected women was approximately 2.5 times higher than those without HIV infection. Colposcopy is indicated particularly in the case of women with a long duration of HIV infection.

## 1. Introduction

Among cancers in women worldwide, cervical cancer ranks fourth for both incidence and mortality following breast, colorectal and lung cancers [1]. The incidence of cervical cancer is still high in developing countries whereas it has decreased significantly in the developed countries over the last several decades. Approximately 80% of cervical cancers occur in developing countries. In Thailand, cervical cancer ranks second after breast cancer with the estimated age-standardized incidence rate of 16.2 per 100,000 women-year. Each year, about 8600 new cases are diagnosed and 5000 women die of cervical cancer [1]. Cervical cancer is currently recognized as the most preventable cancer because almost all cases are caused by human papillomavirus (HPV), the well-known carcinogen [2].

Women infected with human immunodeficiency virus (HIV) have an increased risk of associated infection with HPV, cervical intra-epithelial neoplasia (CIN), and cancer. The prevalence of HPV infection in HIV-infected women is higher than those without HIV infection [3]. Women with HIV have a two to twelve-fold increased risk of developing pre-invasive cervical lesions compared to women without HIV infection [4,5].

Women with low-grade squamous intraepithelial lesion (LSIL) on cytological screening have a 10–30% risk of having a high-grade squamous intraepithelial lesion (HSIL) histology on diagnostic work-up with colposcopy-guided biopsy, whilst the risk of underlying invasive lesion is approximately 1–2% [6].Progression from histologic LSIL to HSIL appears to increase among women with low immunity [7]. According to the American Society of Colposcopy and Cervical Pathology (ASCCP) guidelines, colposcopy is recommended for all HIV-infected women who have LSIL cytology [8]. The data on the risk of underlying high-grade or pre-invasive and invasive cervical lesions in HIV-infected women with LSIL cytology is limited. Accordingly, this study was conducted to evaluate the risk of histological high-grade cervical lesions (CIN2+) in women with HIV infection who had LSIL on cervical cytological screening compared with HIV-uninfected women who had similar cytology.

## 2. Materials and Methods

After approval from the Research Ethics Committee (No. 463/2019), the data of women who had LSIL on cervical cytology and underwent colposcopic examination between January 2008 and December 2019 at Chiang Mai University Hospital (CMUH) were reviewed. The HIV status of every woman was checked before scheduling for colposcopy. Colposcopic examination was performed by the gynecologic oncology fellows under guidance of staff. The inclusion criteria were women aged 18–65 years with LSIL cytology and available histopathology. The exclusion criteria were pregnant women, prior abnormal cervical cytology, history of precancerous lesions and gynecologic malignancies.

Abstracted data included age, parity, menopausal status, current contraceptive use, age at first coitus, number of lifetime sexual partners, smoking status, duration of HIV infection from first positive results to colposcopy, CD4 cell counts within six months of colposcopy, viral load, use of antiretroviral therapy (ART), findings on colposcopy and histopathology on colposcopy-directed biopsy or subsequent cervical conization and hysterectomy.

The sample size of this cross-sectional study was calculated by assuming the risk of having high-grade cervical lesions or invasive lesions defined as cervical intraepithelial neoplasia grade 2 or worse (CIN2+) of 40.9% in HIV-infected women with LSIL cytology and 13.6% in HIV-uninfected women with similar cytology [9]. With a set of α = 0.05 and 1-β = 0.8, the required sample size for HIV-infected cases was 120 women. Women with LSIL cytology who were HIV-negative were recruited for controlled subjects matching 1:1 for the following variables: age (±5 years) and examination time period (±12 months).

Relevant information was exported to Stata version 16 (StataCorp LLC, College Station, TX, USA), which was also used for analysis. Descriptive statistics were used for demographic data. The Fisher’s exact test for categorical variables, *t*-test, or Mann-Whitney U test for continuous variables were used to compare frequencies of variables between the groups. For factors with a *p*-value of less than 0.05 where the missing value did not exceed 25% in univariable analysis, the multivariable analysis using a binary logistic regression model was used to find the independent factors. All statistical tests were two-sided and *p*-value < 0.05 was considered statistically significant.

## 3. Results

A total of 1479 women with LSIL cytology on cervical cancer screening attended the colposcopy clinic at CMUH. Among these, 164 (11.1%) HIV-infected women were identified. Thirty-seven women were excluded because of pregnancy, history of gynecologic malignancies, and unavailable histopathology data. Therefore, 127 women with HIV infection were included for analysis. Clinical characteristics and underlying cervical lesions of the patients with LSIL cytology categorized by HIV Status are shown in Table 1. The mean and median ages of the patients in both groups were 39.02 ± 9.47 and 38.98 ± 9.1 years, respectively, with the interquartile range (IQR) of 32–46 years and a range of 18–61 years. No significant difference was noted between both groups in terms of age, parity, smoking or menopausal status. No contraception was significantly less common in HIV-positive women (21.9% vs. 28.7% in HIV-negative women, *p* = 0.022). Sexual debut before the age of 20 years was significantly higher in HIV-infected women (58.4% vs. 44.2% in HIV-negative women, *p* = 0.005). Multiple sexual partners (≥2) were significantly higher in HIV-infected women compared with that in HIV-uninfected women (82% vs. 43.7%, *p* < 0.001).

The risk of underlying high-grade cervical lesions (CIN2+) in HIV–infected women who had LSIL cytology was significantly higher when compared with that of HIV-negative women who had similar cytology (20.5% versus 9.4%, *p* = 0.021) as shown in Table 1. No invasive cancer was found in both groups. Surprisingly, no CIN2+ occurred in HIV-positive women who did not receive ART. Overall, among 254 women with LSIL cytology, 38 (14.96%) had underlying CIN2+. By univariable and multivariable analyses, the crude odds ratio (OR) was 2.47, 95% CI = 1.18–5.14. After adjustment for the factor of contraception that was significantly different between the two populations and for the missing value did not exceed 25%, the OR of having CIN2+ in HIV-infected women compared with that of HIV-negative women remained significantly higher (adjusted OR = 2.55, 95% CI = 1.11–5.82, *p* = 0.027) as shown in Table 2.

Of the 110 HIV-positive cohort with known duration of HIV infection, women with underlying CIN2+ appeared to have significantly longer duration of HIV infection with the median of 4.5 years compared to 2.0 years in those who had underlying CIN1 or less (*p* = 0.031). Subsequent analysis was carried out to assess the factors affecting underlying high-grade cervical lesions in HIV-infected women as shown in Table 3. The risk of underlying CIN2+ in HIV-positive women was not significantly associated with age, smoking, contraceptive use, CD4 counts or viral load. Women with HIV infection longer than three years had approximately a 3.2-fold increase in the risk of CIN2+ (OR 3.21, 95% CI = 1.24–8.33, *p* = 0.016). The odds of ART on development of high-grade cervical lesions could not be evaluated because no CIN2+ occurred in women who did not receive ART.

## 4. Discussion

In this study, the risk of having histological high-grade cervical lesions or CIN2+ in HIV-infected women who had LSIL on cervical cytological screening was significantly higher than that in HIV-negative women who had similar screening results (20.5% vs. 9.4%, *p* = 0.021). The risk of underlying CIN2+ in HIV-infected women was approximately 2.5 times higher than those without HIV infection. These findings are in line with a previous study on the underlying histopathology of HIV-positive women who had abnormal cervical cytology showing that HIV-infected women had a higher risk of having significant cervical lesions, i.e., CIN2, CIN3 and invasive cancer when compared with HIV-uninfected women irrespective of the severity of abnormal cytological results. HIV-infected women had 2.56 times the risk of underlying CIN2+ [9]. Among women with LSIL cytology, the risk of underlying CIN2+ was significantly higher in HIV-positive women (40.9% vs. 13.6% in HIV-negative women) [9]. In a non-comparison study by Manamela et al. [10]. in 652 HIV-infected women who had LSIL on cervical cytology, it was noted that the risk of underlying CIN2 or worse was high at nearly 60% and suggested that the practice of referral for colposcopy should continue. The lower risk of underlying CIN2+ in HIV-infected women with LSIL cytology in our study (20.5%) compared to the study by Manamela et al. (60%) may result from different baseline characteristics of the studied population, i.e., number of lifetime sexual partners, smoking history, race/ethnicity, and antiretroviral therapy use.

It appeared that the underlying cervical lesions may be more frequently detected in HIV-infected women with mildly abnormal cytological screenings compared to those without HIV infection. In a study of HIV-negative and HIV-positive women evaluated for abnormal cervical cytology with colposcopy-guided biopsy, 49% of the HIV-infected women had cervical lesions more severe than their cytological results, compared with 27% of the HIV-negative patients [11]. In another study of HIV-infected and uninfected women with mild atypia on cytological smears, it was noted that HIV-infected women were approximately four times more likely to have CIN on biopsy, leading to the recommendation that all HIV-positive women with mild cytological atypia undergo colposcopic examination [12].

In contrast, several studies reported no significant effect of HIV infection on underlying cervical lesions. Boardman et al. compared the prevalence of CIN2+ in a cohort of 72 HIV-infected women and 360 HIV-uninfected women with mildly abnormal cervical cytology, i.e., atypical squamous cells of undetermined significance (ASC-US) or LSIL cytology. Both populations were found to be at similar risk for CIN 2+ (15.2% in HIV-positive women compared with 13.3% in HIV-negative women; odds ratio = 1.17, 95% CI = 0.58–2.39). HIV–positive women were as likely as HIV-negative women to have CIN 2 or worse on biopsy [13]. Similarly, Anderson et al. [14]. also noted that underlying cervical lesions in women with ASC-US or squamous intraepithelial lesion cytology did not differ by HIV infection. The difference in the effect of HIV infection on underlying cervical lesions may result from different populations, as our study only included women with LSIL cytology.

In the present study, the incidence of underlying high-grade cervical lesions in HIV-positive and HIV -negative women with LSIL cytology was approximately 15%. No invasive cancer was noted. Such incidence was comparable to the reported incidence of underlying CIN2+ in LSIL cytology in Thailand, which ranged from 11% to 32% [6]. Interestingly, the underlying invasive cancer in women with LSIL cytology in Thailand was quite high at 1.3–1.9% [6]. In comparison, in the region with low incidence of cervical cancer, abnormal cytology of LSIL was associated with CIN2+ in approximately 12–18% of women on colposcopy biopsy and less than 1% had cervical cancer [15,16,17]. Accordingly, referral for colposcopy is recommended for Thai women with LSIL cytology irrespective of HIV status, especially in women who have anxiety about cancer and are expected to default follow-up.

Surprisingly, among 26 HIV-infected women who had high-grade cervical lesions in our study, all occurred in women who received antiretroviral therapy (ART). Accordingly, the odds of ART on development of such lesions could not be evaluated. Data on the effect of ART on underlying high-grade lesions in HIV-positive women are conflicting. Saayman et al. [18] noted that the use of ART had no effect on disease progression, while Manamela et al. [10] found that HIV-infected women who used ART were less likely to have CIN2+ when compared with those who did not use ART. However, a systematic review and meta-analysis on the effects of ART on HPV acquisition and cervical neoplasia in HIV-infected women showed that ART was associated with a significantly decreased risk of CIN2+ incidence and CIN progression and a significantly increased likelihood of CIN regression. In addition, ART was also significantly associated with a reduction in invasive cervical cancer incidence. Early ART initiation and sustained adherence appear to reduce the incidence and progression of CIN and ultimately reduce the incidence of invasive cervical cancer in women with HIV infection [19]. ART utilization can inhibit the proliferation of HIV and indirectly maintain the serum CD4 T-lymphocyte level. Although the effects of ART on the development of CIN have not been fully elucidated, the currently available clinical evidence suggests that ART may prevent cervical carcinogenesis by enhancing the immune surveillance function and increasing the likelihood of clearance of HPV infection [19].

Increasing immunosuppression in HIV-positive women with LSIL cytology in our study was not significantly associated with underlying CIN2+. However, the lack of statistical significance in our study may result from the small sample size. Anderson et al. [14] also noted that abnormal histologic findings did not differ by CD4 cell counts in HIV-positive women with mildly cytological abnormalities. In contrast, other investigators found that low CD4 cell counts were significantly associated with development of high-grade cervical lesions [20,21]. The study by Zhang et al. on the association of cervical neoplastic lesions with CD4 cell counts and the use of ART among HIV-infected women found that the prevalence of cervical neoplasia appeared to be higher in women who had lower CD4 cell counts. The authors concluded that high CD4 cell counts might have protective effects against the development of cervical neoplastic lesions [22]. Low CD4 cell counts in HIV-infected women indicate the existence of immunosuppression, which can be associated with an increased risk of cervical neoplasia.

HIV-positive women with underlying CIN2+ in the present study had significantly longer duration of HIV infection with the median of 4.5 years compared to 2.0 years in those who had underlying CIN1 or less (*p* = 0.031). Women infected with HIV longer than three years had an approximately 3.2-fold increase in the risk of underlying CIN2+ (OR 3.21, 95% CI = 1.24–8.33, *p* = 0.016). Duration of HIV infection might be associated with immune homeostasis and ART resistance in women with HIV infection [23]. However, several studies noted that duration of HIV infection was not significantly associated with development of high-grade cervical lesions. The study by Brito et al. [21] showed that there was no significant difference in the duration of HIV infection between women who had low-grade and high-grade cervical diseases (median duration = 6.9 years and 6.4 years, respectively). However, such results might be limited by the small number of 20 HIV-positive women in each group. A population-based study among 545 HIV-infected women in China reported no significant interaction between CIN development and duration of HIV infection [22].

The retrospective nature of our study imposes inherent limitations, e.g., inconsistent collecting and recording of the patients’ characteristics. Several potential factors expected to be associated with an increased risk of high-grade cervical lesions were not available in the entire HIV-positive cohort, such as viral load, CD4 cell counts and the use of ART. This may result in inability to evaluate the significance or lacking statistical significance of the comparison. The data regarding HPV status affecting development of high-grade cervical lesions were not available in both cohorts. Certain characteristics of both the HIV- infected and HIV- uninfected populations may also limit the generalizability of the study. Moreover, no data on long-term follow-up were available to evaluate the future risk of developing CIN2+ in HIV-infected and HIV-uninfected women who had LSIL cytology. The strengths of our study include the size of the HIV-infected cohort and the study design in which HIV-uninfected women with LSIL cytology were recruited from a large colposcopy database as controlled subjects by matching 1:1 for age (±5 years) and examination time period (±12 months). In addition, all patients in this study were treated at a single institution with the same management protocol, and all the pathological specimens were examined by expert gynecologic pathologists.

## 5. Conclusions

The risk of underlying high-grade cervical lesions in HIV-infected women with LSIL cytology was approximately 20% or 2.5 times higher than that in HIV-uninfected women with similar cytology. Referral for colposcopy is recommended to detect and treat precancerous cervical lesions in these patients, particularly women with a long duration of HIV infection.

## Figures and Tables

**Table 1 ijerph-18-10211-t001:** Characteristics and underlying cervical lesions of the patients with LSIL. Cytology categorized by HIV status (n = 254).

Characteristics	HIV		*p*-Value
	Positive (n = 127)	Negative (n = 127)	
Age (years)			
Mean ± SD	39.02 ± 9.47	38.98 ± 9.1	0.973
Median (IQR)	39 (32–46)	39 (32–46)	
Parity			0.127
0	30 (23.6)	38 (30.0)	
1	61 (48.0)	45 (35.4)	
≥2	36 (28.4)	44 (34.6)	
Smoking (n = 199)			0.092
No	96 (89.7)	89 (96.7)	
Yes	11 (10.3)	3 (3.3)	
Menopause			0.160
No	109 (85.8)	117 (92.1)	
Yes	18 (14.2)	10 (7.9)	
Contraception (n = 197)			0.022
None	21 (21.9)	29 (28.7)	
Hormone	22 (22.9)	38 (37.6)	
Condom	25 (26.0)	15 (14.9)	
Others	28 (29.2)	19 (18.8)	
Age at first sexual intercourse (n = 175)			0.005
<20 years	52 (58.4)	38 (44.2)	
≥20 years	37 (41.6)	48 (55.8)	
Median (IQR)	19 (17–21)	20 (18–23)	0.070
Number of sexual partners (n = 176)			<0.001
0	0 (0)	1 (1.1)	
1	16 (18)	48 (55.2)	
≥2	73 (82)	38 (43.7)	
Underlying lesions			0.021
≤CIN1	101 (79.5)	115 (90.6)	
CIN2+	26 (20.5)	12 (9.4)	

LSIL, low-grade squamous intraepithelial lesion. HIV, human immunodeficiency virus. CIN, cervical intraepithelial neoplasia. IQR, interquartile range.

**Table 2 ijerph-18-10211-t002:** Underlying CIN2+ in women with LSIL cytology by HIV Status.

Underlying Lesions	HIV		Odds Ratio (95% CI)		*p*-Value
	Positive (%)	Negative (%)	Crude	Adjusted	
≤CIN1	101 (79.5)	115 (90.6)	1	1	
CIN2+	26 (20.5)	12 (9.4)	2.47 (1.18–5.14)	2.55 (1.11–5.82)	0.027

CI, confidence interval. CIN, cervical intraepithelial neoplasia.

**Table 3 ijerph-18-10211-t003:** Risk of underlying CIN2+ in women with HIV infection.

Variables	Odds Ratio (95% CI)	*p*-Value
Age (≥30 vs. <30 years) (n = 127)	0.51 (0.18–1.41)	0.196
Smoking vs. not smoking (n = 107)	0.96 (0.19–4.84)	0.963
Contraception		
Condom vs. no condom (n = 96)	0.26 (0.05–1.20)	0.083
Hormone vs. no hormone (n = 96)	1.61 (0.53–4.85)	0.400
Hormone vs. condom (n = 68)	2.50 (0.70–8.92)	0.158
CD4 count (<200 vs. ≥200 cells/mL) (n = 107)	0.73 (0.24–2.19)	0.574
Viral load (≥40 vs. <40) (n = 63)	1.89 (0.51–6.94)	0.339
HIV infection duration, (≥3 vs. <3 years) (n = 110)	3.21 (1.24–8.33)	0.016

CI, confidence interval. CD4, cluster of differentiation 4. HIV, human immunodeficiency virus.

## Data Availability

The datasets analyzed during this study are available from the corresponding author upon reasonable request.

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
