# Peer review of "HIV-Infected Women with Low-Grade Squamous Intraepithelial Lesion on Cervical Cytology Have Higher Risk of Underlying High-Grade Cervical Intraepithelial Neoplasia"

_ijerph, 2021, doi:10.3390/ijerph181910211_

Round 1

Reviewer 1 Report

well written manuscript and ready for publication in my view 

I did make some suggestions about limitations of the study that the authors may elaborate on if relevant 

Author Response

Response to Reviewer 1

  1. Suggestions about limitations of the study that the authors may elaborate on if relevant: suggest more limitations of the study for discussion, eg. age of sexual debut and multiple sexual partners may independently from HIV infection and HPV also have an influence on neoplastic potential

Response: Both younger age of sexual debut and multiple sexual partners were significantly more common in HIV-positive cohort despite small population, accordingly both factors were not described in the limitations of the study. The sentence “The data of HPV status affecting development of high-grade cervical lesions were not available in both cohorts” was added in the limitations of the study. 

  1. ART or lack thereof may lead to earlier death and thus masking effect on prevalence of cervical neoplasia

Response: We have already mentioned that the data of ART were not available in the entire HIV-positive cohort which is one of the limitations of our study.

Reviewer 2 Report

the paper is overall well written and interesting. the findings are expected but sound. Authors should update bibliography

Author Response

Response to Reviewer 2

  1. Authors should update bibliography

Response: From literature review, we found that few studies on the prevalence of high-grade cervical lesions in HIV-positive women with abnormal cervical cytology were reported after 2010.

Reviewer 3 Report

This is a study about LSIL and CIN2+ in HIV-postive patients.

The sample is small and this is discussed.

I think that one ore two graphics in the results may help the reader.

The discussion section should be shortened.

Author Response

Response to Reviewer 3

  1. One or two graphics in the results may help the reader.

Response: The significant findings of the study have been clearly described in the Results section, accordingly the graphic presentation is not necessary.

  1. The discussion section should be shortened.

Response: There are several factors significantly associated with underlying high-grade cervical lesions in HIV-positive women that have to be discussed. Accordingly, the discussion part was quite a little bit longer than usual for better understanding.

Reviewer 4 Report

This research article presents a case-control study examining the rate of cervical neoplasia in HIV-infected women compared to HIV-negative controls from a single medical center in northern Thailand from 2008 to 2019. A higher odds ratio for CIN2+ was found for HIV positive patients. This supported the conclusion that colposcopy is indicated particularly in the case of women with a long duration of HIV infection.

While the study appears meticulously designed, it seems to be generally in agreement with similar studies such as those discussed in the conclusion. Specifically, there do not appear to be any changes to clinical practice based on this study. It might help to discuss what further directions are indicated by these data and conclusions.

The abstract would be improved by omitting non-significant results and including the clinical practice recommendations from the conclusion.

Line 99-100 suggests that “HIV-positive women were significantly less likely to  use  contraceptive  methods  (21.9  %  vs  28.7  %  in  HIV-negative  women,  P  = 0.022).” However, this seems to be the inverse of the data presented in table 1 where HIV-positive women report 21.9% have “none” for contraceptive use vs 28.7 in the HIV-negative group. Please clarify.

Author Response

Response to Reviewer 4

  1. There do not appear to be any changes to clinical practice based on this study. It might help to discuss what further directions are indicated by these data and conclusions.

Response: Currently, HPV testing combined with cervical cytology is used for screening. The result of HPV testing will certainly influence clinical practice in HIV-positive women.

  1. The abstract would be improved by omitting non-significant results and including the clinical practice recommendations from the conclusion.

Response:  Non-significant results in the abstract “Mean ages ± SD of the HIV-positive and HIV-negative groups were 39.02±9.47 years and 38.98±9.1 years, respectively”   was deleted. The sentence “Colposcopy is indicated particularly in the case of women with a long duration of HIV infection” was added in the Conclusion of the Abstract.

  1. Line 99-100 suggests that “HIV-positive women were significantly less likely to use  contraceptive  methods  (21.9  %  vs  28.7  %  in  HIV-negative  women,  P  = 0.022).” However, this seems to be the inverse of the data presented in table 1 where HIV-positive women report 21.9% have “none” for contraceptive use vs 28.7 in the HIV-negative group. Please clarify.

Response: We change the sentence to “No contraception was significantly less common in HIV-positive women (21.9  %  vs  28.7  %  in  HIV-negative  women,  P  = 0.022)” to clarify the result.